# High-efficiency gold recovery by additive-induced supramolecular polymerization of β-cyclodextrin

Huang Wu [1,6], Yu Wang [1,6], Chun Tang[1], Leighton O. Jones [1], Bo Song [1], Xiao-Yang Chen [1], Long Zhang [1], Yong Wu[1], Charlotte L. Stern [1], George C. Schatz [1], Wenqi Liu [2] ✉ & J. Fraser Stoddart [1,3,4,5] ✉

Developing an eco-friendly, efficient, and highly selective gold-recovery technology is urgently needed in order to maintain sustainable environments and improve the utilization of resources. Here we report an additive-induced gold recovery paradigm based on precisely controlling the reciprocal transformation and instantaneous assembly of the second-sphere coordinated adducts formed between β-cyclodextrin and tetrabromoaurate anions. The additives initiate a rapid assembly process by co-occupying the binding cavity of β-cyclodextrin along with the tetrabromoaurate anions, leading to the formation of supramolecular polymers that precipitate from aqueous solutions as cocrystals. The efficiency of gold recovery reaches 99.8% when dibutyl carbitol is deployed as the additive. This cocrystallization is highly selective for square-planar tetrabromoaurate anions. In a laboratory-scale gold-recovery protocol, over 94% of gold in electronic waste was recovered at gold concentrations as low as 9.3 ppm. This simple protocol constitutes a promising paradigm for the sustainable recovery of gold, featuring reduced energy consumption, low cost inputs, and the avoidance of environmental pollution.

Gold, an indispensable element in human society from time immemorial, is widely used in currency and jewelry manufacture[1], electronic fabrication[2], medicine production[3], and chemical synthesis[4]. Gold mining, however, is known notoriously to be one of the most environmentally destructive industries in today's world. Massive amounts of cyanide[5] and mercury[6] are used to extract gold from ores each year, resulting in enormous waste streams contaminated with lethal cyanide and heavy metals, along with colossal amounts of carbon emissions and excessive energy consumption. In order to develop sustainable technologies for gold production and recovery, many alternative methods[7], based upon the selective extraction or adsorption of gold from leaching solutions, have been developed. These methods include leaching electronic waste (e-waste) and gold ores with a single organic[8–12]/inorganic[13] extraction reagent, or specific combinations[14,15] of extraction reagents and organic solvents, not to mention the adsorption of ionic gold complexes with metal-organic frameworks[16,17] and polymers[18–20]. As an alternative approach to extraction and adsorption, selective co-precipitation[21–24] based on second-sphere coordination[25,26] have proven increasingly popular for metal separations given their significant advantages—e.g., simple operation, ease of industrialization, minimal energy consumption, and zero hazardous emissions.

First-sphere coordination[27], advanced in the early part of the 20th century by Nobel Laureate in Chemistry, Alfred Werner, refers

[1]Department of Chemistry, Northwestern University, 2145 Sheridan Road, Evanston, IL 60208, USA. [2]Department of Chemistry, University of South Florida, 4202 East Fowler Avenue, Tampa, FL 33620, USA. [3]School of Chemistry, University of New South Wales, Sydney, NSW 2052, Australia. [4]Department of Chemistry, Stoddart Institute of Molecular Science, Zhejiang University, 310027 Hangzhou, China. [5]ZJU-Hangzhou Global Scientific and Technological Innovation Center, 311215 Hangzhou, China. [6]These authors contributed equally: Huang Wu, Yu Wang. ✉e-mail: wenqi@usf.edu; stoddart@northwestern.edu

to the coordinative bonding interactions between first-coordination sphere ligands and transition metals. Under the umbrella of supramolecular[28,29] and host−guest[30,31] chemistry, the investigation of second-sphere coordination[32–34], which involves the noncovalent interactions between the first-sphere ligand and a macrocyclic molecule as the second-sphere ligand, has skyrocketed during the past several decades. In this context, many well-crafted macrocyclic receptors, e.g., crown ethers[32], cyclodextrins[35,36], calixarenes[37], cucurbiturils[38], and others[39,40] have emerged as promising second-sphere coordination ligands, enabling the modulation of the chemical and physical properties of transition metal complexes. These macrocycles exhibit highly specific recognition for particular metal cationic complexes, including those containing Rh[+41], Ru[2+42], Gd[3+43], and Yb[3+44], as well as serve as anion receptors[45–47] for negatively charged metal complexes, such as $[ReO_4]^{-48}$, $[CdCl_4]^{2-49}$, $[PtCl_6]^{2-50}$, polyoxometalates[51] and others[52]. The precise control of the assembly and reciprocal transformation of these second-sphere-coordinated adducts, however, remains challenging. Some of the second-sphere-coordinated adducts exhibit[53] unique crystallinity, a property that paves the way for using second-sphere coordination to recycle precious metals from e-waste. Employing this protocol, we have separated[54,55] gold from ores where α-cyclodextrin acts preferentially as a second-sphere coordinator for hydrated potassium tetrabromoaurate. When it comes to practical gold recovery, this protocol, however, still suffers from several limitations, including the fact that (i) a high content of gold ($[KAuBr_4] > 6$ mM) in the leaching solution is required, (ii) additional potassium ions are mandatory, (iii) a high concentration of acid in the leaching solution prevents the formation of co-precipitates, (iv) gold-recovery efficiency is below 80% when performed at room temperature, and (v) the cost of α-cyclodextrin is relatively high. Hence, the development of more efficient and economic gold separation technology aligned with practical gold recovery is significant and necessary.

Herein, we demonstrate an additive-induced gold separation paradigm based on precisely controlling the reciprocal transformation of the second-sphere-coordinated adducts formed between β-CD and $[AuBr_4]^-$ anions. Mechanistic investigations reveal that the additives drive rapid assembly of β-CD and $[AuBr_4]^-$ anions by forcing the $[AuBr_4]^-$ anions to move from the inner cavity to the primary faces of two β-CD tori, while the additives occupy the space between the secondary faces of the two β-CD tori, thus forming infinite one-dimensional supramolecular polymers that precipitate from aqueous solutions as cocrystals. A wide range of common organic solvents can be employed as additives. Solvents with high boiling points provide higher gold-recovery efficiencies. A gold-recovery efficiency of 99.8% has been achieved when dibutyl carbitol is used as the additive. The rapid cocrystallization is highly selective for $[AuBr_4]^-$ anions, while no precipitate is observed when using metal cations along with other structurally similar anions, such as $[PdBr_4]^{2-}$ and $[PtBr_4]^{2-}$. A laboratory-scale gold-recovery protocol, aligned with an attractive strategy for the practical recovery of gold metal, has been established, wherein 94% of gold is recovered directly from a leaching solution of gold-bearing scrap at gold concentrations as low as 9.3 ppm. In principle, this highly selective and fast cocrystallization procedure can be used for gold recovery from gold-bearing ores and e-waste.

## Results

### Encapsulation KAuBr₄ by β-cyclodextrin

Second-sphere coordination (Fig. 1a) of KAuBr₄ with β-CD in aqueous solutions was investigated by [1]H NMR spectroscopy. After adding an excess of KAuBr₄ into a D₂O solution of β-CD, all the protons on the β-CD show (Fig. 1b) notable changes in their chemical shifts. The resonances for protons H-1 and H-2 on the D-glucopyranosyl residues of β-CD show upfield shifts ($\Delta\delta = -0.03$ and $-0.03$ ppm for H-1 and H-2), while the resonances for protons H-3, H-4, and H-5 on the

D-glucopyranosyl residues undergo downfield shifts ($\Delta\delta = 0.05$, 0.03 and 0.33 ppm for H-3, H-4, and H-5, respectively). Proton H-5 exhibits the largest downfield shift, while the resonance for proton H-6 separates into two sets of peaks, indicating that the $[AuBr_4]^-$ anions are located near the primary face of β-CD. The binding affinity between β-CD and KAuBr₄ in D₂O was determined (Fig. 1c and Supplementary Figs. 10, 11) by [1]H NMR titration. By tracking the changes in the chemical shift of H-5, the binding constant ($K_a$) between the $[AuBr_4]^-$ anion and β-CD was determined (Fig. 1d) to be $4.47 \times 10^4$ M$^{-1}$. The corresponding $\Delta G°$ value was calculated to be $-6.3$ kcal mol$^{-1}$. Additionally, a strong peak with an $m/z$ value of 1651.0059 was observed (Supplementary Fig. 18) in the high-resolution mass spectrum, corresponding to the 1:1 complex $[[AuBr_4]^- \subset β-CD]^-$. Its isotopic pattern matches (Fig. 1e) well with the theoretical one, strongly supporting the formation of the host−guest complex in aqueous solutions.

The solid-state superstructure of the 1:1 complex was determined unambiguously by single-crystal X-ray diffraction analysis. Brown single crystals were obtained by slowly cooling an aqueous solution of KAuBr₄ and β-CD from 90 °C to room temperature over 6 h. The $[AuBr_4]^-$ anion, which is encapsulated (Fig. 1f) inside the cavity of β-CD, is located closer to the primary face (1°) of β-CD than to its secondary face (2°). The β-CD tori are distorted (Fig. 1f) elliptically and elongated along the $[AuBr_4]^-$ plane. The lengths of the major and minor axes for the distorted β-CD tori are 13.9 and 12.6 Å, respectively. In the superstructure of the $[AuBr_4]^- \subset β-CD$ complex, five of the H-5 protons on β-CD interact (Supplementary Fig. 1) closely with bromine atoms. The [C−H···Br−Au] distances (Supplementary Table 1) range from 3.0 to 3.4 Å. It is for this reason that the H-5 protons show the largest change in their chemical shift in the [1]H NMR spectra. Three of the H-6 protons on β-CD also have close contact with the bromine atoms in the anion. The [C−H···Br−Au] distances (Supplementary Table 1) are 3.2−3.4 Å. These observations suggest that the $[AuBr_4]^- \subset β-CD$ complex is stabilized by multiple weak [C−H···Br−Au] hydrogen-bonding interactions, aided and abetted by a hydrophobic effect overall. These noncovalent interactions were visualized (Fig. 1g and Supplementary Fig. 37) by an independent gradient model (IGM) analysis. All the results confirm the formation of second-sphere-coordinated KAuBr₄⊂β-CD adducts in the solution and solid states.

### Additive-induced supramolecular polymerization

Upon adding a trace amount of common organic solvents, which serve as additives, to a 1 M HBr aqueous solution of the KAuBr₄⊂β-CD complex, brown co-precipitates form immediately. (Fig. 2a and Supplementary Movie 1). On the contrary, after mixing the same additives and β-CD with aqueous solutions of K₂PdBr₄ or K₂PtBr₄, which have lower binding affinities ($1.45 \times 10^2$ and 33.3 M$^{-1}$, respectively, Supplementary Figs. 12−15) with β-CD, no precipitation is observed (Fig. 2a). This observation establishes the fact that a trace amount of organic solvent can induce selective co-precipitation of the $[AuBr_4]^- \subset β-CD$ complex, laying the foundation for developing a simple and effective gold-recovery technology. In order to verify the generality of this additive-induced co-precipitation behavior, various organic solvents and oils were added (Fig. 2b) to aqueous solutions of the KAuBr₄⊂β-CD complex. Dibutyl carbitol (DBC), isopropyl ether ($i$Pr₂O), hexane, dichloromethane (CH₂Cl₂), chloroform (CHCl₃), benzene and toluene can induce co-precipitation, while no precipitate was observed when diethyl ether, ethyl acetate, and other oils were used as additives. In order to quantify the yield of gold co-precipitates, UV−Vis absorption spectroscopy was performed. The characteristic absorption peaks for $[AuBr_4]^-$ at 381 and 253 nm were used (Supplementary Figs. 19, 20) to determine the concentration of $[AuBr_4]^-$ in the filtrates. The efficiencies of gold recovery for different samples were calculated based on the initial and residual concentrations of the $[AuBr_4]^-$ anions in the aqueous solutions. We observed gold-recovery efficiencies ranging (Fig. 2b) from 27.0% to 99.8%, depending on the additives. It is worth

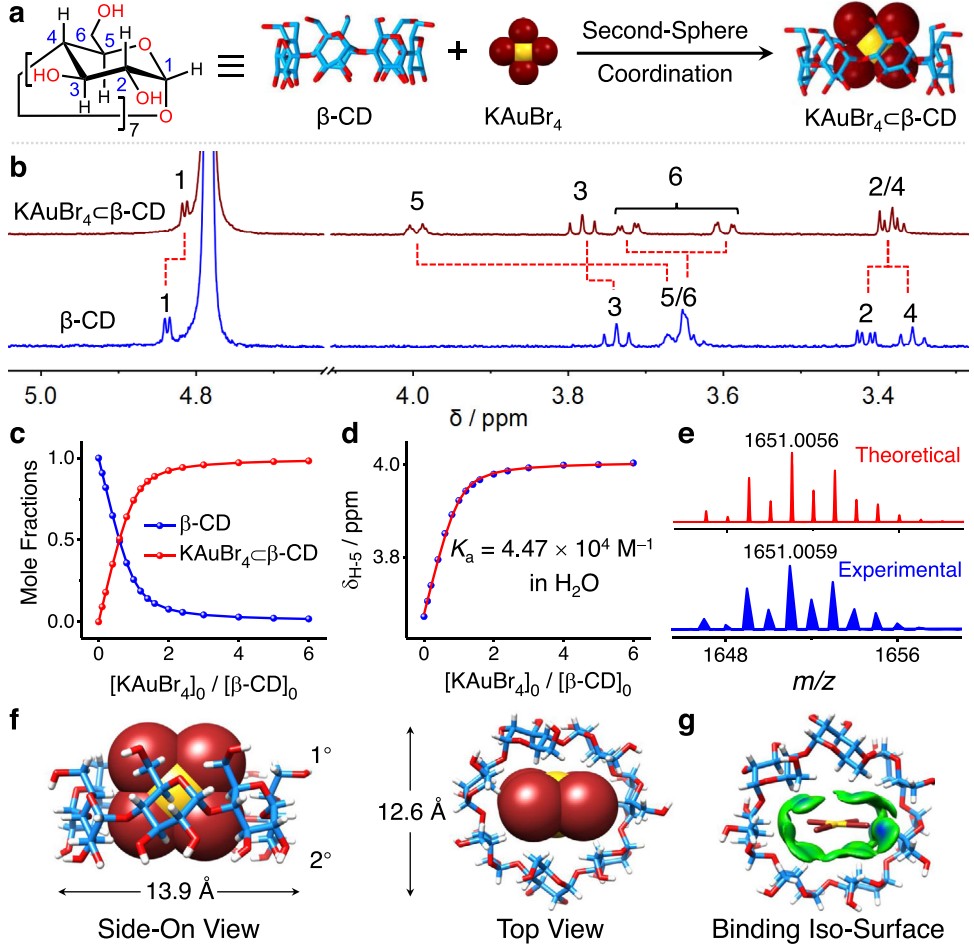

**Fig. 1 | Host−guest interactions between β-CD and KAuBr$_4$. a** Schematic illustration of the recognition of [AuBr$_4$]$^-$ anions facilitated by second-sphere coordination with β-CD. **b** $^1$H NMR Spectra (600 MHz, D$_2$O containing 0.5 M DBr, [β-CD] = 2.5 × 10$^{-4}$ M, [KAuBr$_4$] = 1.5 × 10$^{-3}$ M, 298 K) of β-CD and KAuBr$_4$⊂β-CD. **c** Changes in mole fractions for β-CD (blue trace) and KAuBr$_4$⊂β-CD (red trace) as a function of the KAuBr$_4$-to-β-CD ratio during the $^1$H NMR titration. **d** $^1$H NMR Titration isotherm created by monitoring changes in the chemical shift of H-5 upon adding 0−6 molar equivalents of KAuBr$_4$ to an aqueous solution of β-CD. **e** Theoretical and experimental high-resolution mass spectra of [AuBr$_4$]$^-$⊂β-CD in aqueous solutions. **f** The solid-state superstructure of [AuBr$_4$]$^-$⊂β-CD. **g** Intermolecular binding iso-surface of [AuBr$_4$]$^-$⊂β-CD. Solvent molecules in crystal superstructure have been omitted for the sake of clarity. The "1°" represents the primary face and the "2°" represents the secondary face of β-CD. H white, C skyblue, O red, Br brown, Au yellow.

noting that when DBC was used as the additive, the [AuBr$_4$]$^-$ anions were almost entirely precipitated from aqueous solutions with a gold-recovery efficiency of 99.8%. This efficiency is much higher than that (78.3%) obtained in a previous method[54] using α-CD. Gold-recovery efficiency was also optimized with respect to the amount of the additive. The results indicate that 0.1% (v/v) of DBC enables (Fig. 2c) a high gold precipitation yield of 99.7%. These observations suggest that DBC is the best candidate for gold recovery among all additives tested.

Filtration of the suspension, which was obtained upon the addition of DBC to an aqueous solution of the KAuBr$_4$⊂β-CD complex, led to the isolation of a brown solid. The Fourier transform infrared (FTIR) spectrum of this solid reveals (Fig. 2d) sharp vibrational bands for the KAuBr$_4$⊂β-CD complex at 1021 and 1152 cm$^{-1}$, in addition to the characteristic vibration bands for DBC at 2864 and 2925 cm$^{-1}$. These data confirm the formation of a ternary adduct between β-CD, KAuBr$_4$, and DBC−namely, KAuBr$_4$·DBC⊂2β-CD. In order to investigate the crystallinity of the KAuBr$_4$·DBC⊂2β-CD adduct, powder X-ray diffraction (PXRD) analysis was performed. The PXRD pattern shows (Fig. 2e) a series of sharp diffraction peaks, indicating that the co-precipitate is a highly crystalline material. In the scanning electron microscopic (SEM) images (Fig. 2f, g) of the air-dried KAuBr$_4$·DBC⊂2β-CD suspension, a plethora of angular rod-like microstructures with diameters in the range of several micrometers and lengths up to hundreds of

micrometers were observed. SEM-equipped energy-dispersive X-ray spectroscopic (SEM-EDS) elemental maps uncover (Supplementary Fig. 34) a homogeneous distribution of the elements of carbon, oxygen, bromine, and gold throughout the microrods, confirming the formation of KAuBr$_4$·DBC⊂2β-CD adducts. The thermogravimetric analysis (TGA) profile for the KAuBr$_4$·DBC⊂2β-CD co-precipitate reveals (Supplementary Fig. 51a) that it begins to suffer mass loss at temperatures around 100 °C, most likely because of the loss of water of crystallization. Significant decomposition occurs around 160 and 280 °C, arising from halide release and the breakdown of β-CD. Finally, over 70 wt% of the original mass of the co-precipitate was lost at 800 °C. The TGA traces for the co-precipitates obtained by adding other additives behaved in a similar fashion (Supplementary Fig. 51b) to that of the KAuBr$_4$·DBC⊂2β-CD adduct over the temperature range from 40 to 800 °C, indicating they have similar components.

**Mechanism of additive-induced supramolecular polymerization**
The mechanism for DBC-induced supramolecular polymerization has been elucidated by X-ray crystallography. Brown single crystals of the ternary complex formed between KAuBr$_4$, β-CD, and DBC were obtained by slow vapor diffusion of DBC into an aqueous solution of KAuBr$_4$ and β-CD over 3 days. Single-crystal X-ray diffraction analysis reveals that the [AuBr$_4$]$^-$ anion is centered (Fig. 3a) between the

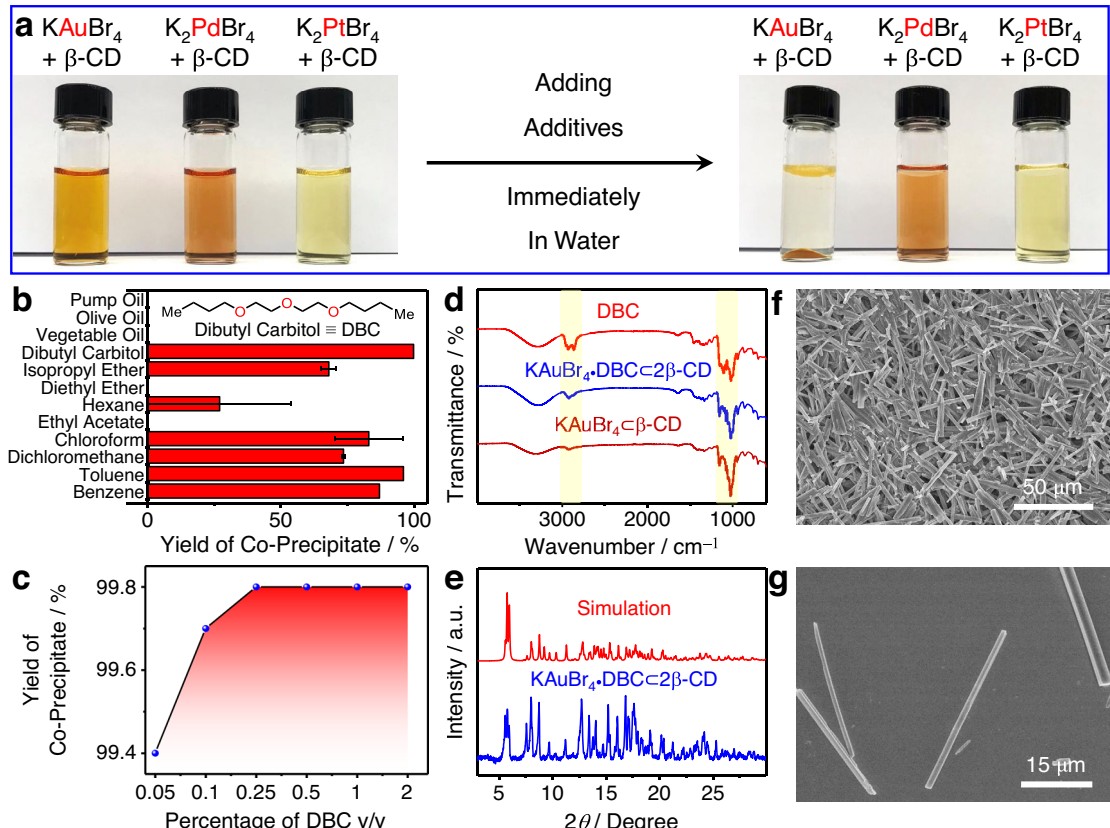

**Fig. 2 | Additive-induced supramolecular polymerization between β-CD, KAuBr₄, and a collection of additives. a** Visual display of the selective co-precipitation between β-CD and KAuBr₄ upon adding additives. **b** Effect of adding different additives on the yield of gold-bearing co-precipitates. The error bar represents the standard deviation of two independent measurements. **c** Effect of the amount of dibutyl carbitol (DBC) on the yield of gold-bearing co-precipitates. **d** FTIR Spectra of DBC, KAuBr₄⊂β-CD, and the KAuBr₄·DBC⊂2β-CD co-precipitate obtained by adding DBC to an aqueous solution of KAuBr₄⊂β-CD complex. The highlighted regions with yellow show their characteristic vibration peaks. **e** Powder X-ray diffraction patterns of the KAuBr₄·DBC⊂2β-CD co-precipitate, compared with a simulated pattern derived from X-ray crystallographic data for HAuBr₄·DBC⊂2β-CD cocrystal. **f, g** SEM Images of the KAuBr₄·DBC⊂2β-CD microcrystals prepared by adding DBC to an aqueous solution of KAuBr₄⊂β-CD.

primary faces of two adjacent β-CD tori. The H-5 protons on the primary faces of the two neighboring β-CD tori are close (Fig. 3b and Supplementary Fig. 3) to the four bromine atoms in the [AuBr₄]⁻ anion with [C−H···Br−Au] contacts ranging from 3.2 to 3.5 Å (Supplementary Table 2). All the H-6 protons on the two β-CD tori are in close contact with bromine atoms. The [C−H···Br−Au] distances range from 3.0 to 3.3 Å (Supplementary Table 2). The multiple [C−H···Br−Au] hydrogen-bonding interactions between the inward-facing H-5 and H-6 protons and the four bromine atoms constitute the major interactions anchoring the [AuBr₄]⁻ anions. Additionally, the bromine atoms in the [AuBr₄]⁻ interact (Fig. 3b) with DBC on account of a [C−H···Br−Au] hydrogen bond with a distance (Supplementary Fig. 3d) of 3.0 Å, while DBC occupies (Fig. 3a) the internal cavities of two neighboring β-CD tori. With the DBC serving as a connector and multiple intermolecular hydrogen bonds between the secondary faces, the two β-CD tori adopt (Fig. 3b) a head-to-head packing arrangement, forming a supramolecular dimer. Because the length (18.7 Å) of stretched-out DBC is much longer than the depth (13.2 Å) of the β-CD dimer, DBC adopts (Fig. 3a) a folded conformation inside the internal cavity of the dimer in order to maximize surface contacts. The folded DBC is sustained (Supplementary Fig. 3c) by two sets of [H−O···H−C] hydrogen bonds between the oxygen atoms on the secondary faces of the β-CD tori and the methylene hydrogens on DBC, as well as multiple Van der Waals interactions between inward-facing H-3 and H-5 protons on the β-CD tori and the carbon and hydrogen atoms in DBC. The two [H−O···H−C] distances were found (Supplementary Fig. 3c) to be 2.6 and 2.7 Å, while the distances associated with the Van der Waals interactions range

(Supplementary Fig. 3c) from 2.1 to 2.9 Å. The β-CD tori undergo (Fig. 3a) elliptical deformations to accommodate the folded conformations of DBC, where the lengths of the major and minor axes for the distorted β-CD tori are 13.8 and 12.9 Å, respectively. It is worth mentioning that K⁺ ions are absent (Supplementary Fig. 3) in the crystal superstructures. A possible explanation is that protons replace K⁺ ions during crystallization. With the [AuBr₄]⁻ anions and DBC molecules as connectors, the β-CD tori are arranged (Fig. 3c) in a head-to-head and tail-to-tail manner extending along the *c* axis, forming an infinite one-dimensional (1D) supramolecular polymer. Bundles of these polymers are tightly packed as a result of intermolecular hydrogen bonds between the 1D columns, forming needle-like single crystals. The simulated PXRD pattern, derived from the single-crystal X-ray crystallographic data of HAuBr₄·DBC⊂2β-CD, matches (Fig. 2e) well with the experimental one, indicating that the superstructure of the HAuBr₄·DBC⊂2β-CD microcrystals obtained by solution-phase synthesis is consistent with its single-crystal X-ray diffraction analysis.

Based on the solid-state superstructures of [AuBr₄]⁻ and β-CD, obtained before and after adding DBC, we propose the mechanism (Fig. 3d) of additive-induced rapid cocrystallization and concomitant co-precipitation. In the aqueous solution of [AuBr₄]⁻ and β-CD, the primary species are the 1:1 [AuBr₄]⁻⊂β-CD complex and free β-CD. After adding the hydrophobic DBC, it serves as an additional guest by co-occupying the binding cavity of two β-CD tori and forces the [AuBr₄]⁻ anions to move to the primary faces of the β-CD tori, forming a unique heterodimeric encapsulation complex. With DBC connecting the two secondary faces and [AuBr₄]⁻ anions linking two primary faces

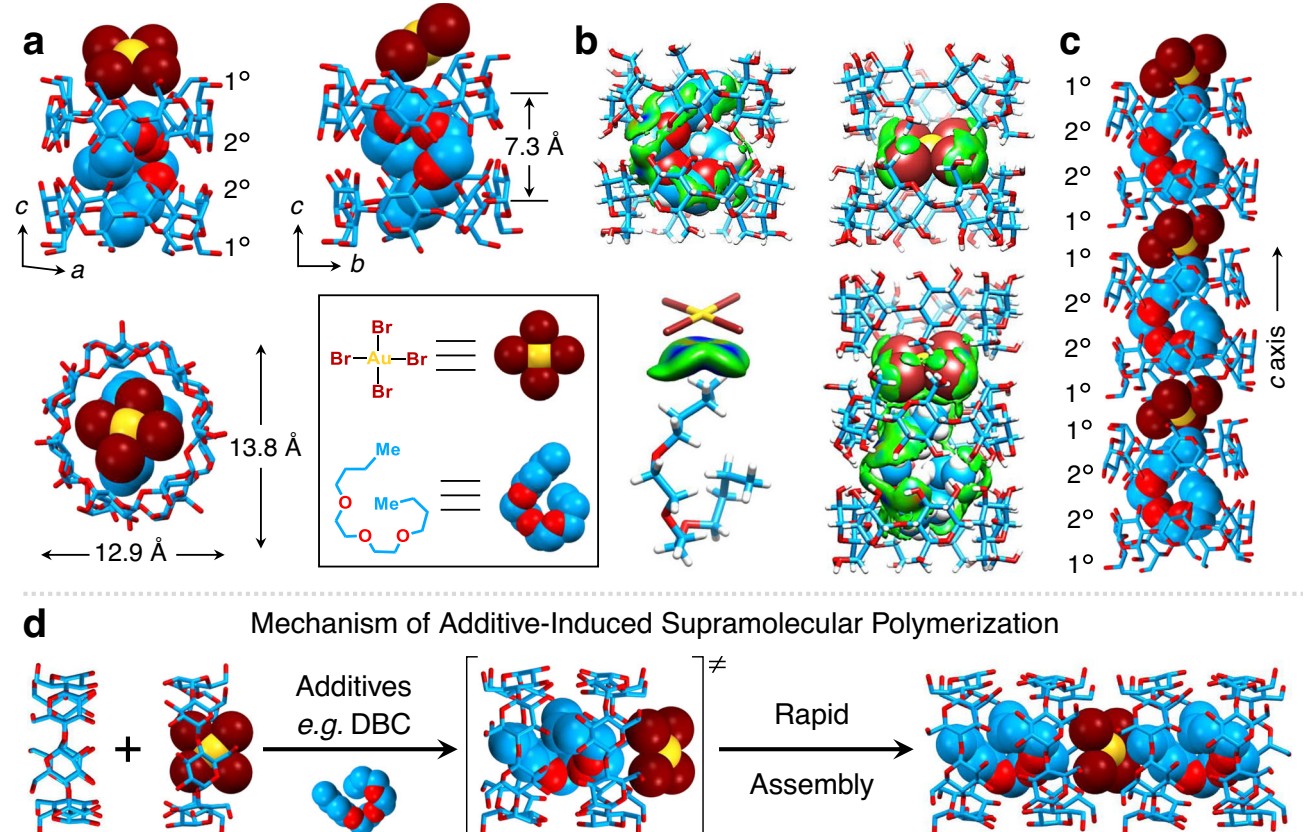

**Fig. 3 | Solid-state superstructure and intermolecular binding iso-surface of the HAuBr₄·DBC⊂2β-CD cocrystal.** It illustrates the mechanism of additive-induced supramolecular polymerization. **a** Capped-stick and space-filling representation of different views of the HAuBr₄·DBC⊂2β-CD quaternary complex, showing the packing mode and dimensions of the complex. **b** Intermolecular binding iso-surface between β-CD and DBC, β-CD and [AuBr₄]⁻, DBC and [AuBr₄]⁻, respectively. **c** Capped-stick and space-filling representation of the one-dimensional nanostructure extending along the *c*-axis, in which the β-CD tori form a continuous channel occupied by alternating DBC and [AuBr₄]⁻ anions. Solvent molecules have been omitted for the sake of clarity. The "1°" represents the primary face and the "2°" represents the secondary face of β-CD. H white, C skyblue, O red, Br brown, Au yellow. **d** Schematic illustration of the mechanism of additive-induced supramolecular polymerization after adding various additives to the solution of β-CD and [AuBr₄]⁻ anions.

of the β-CD tori, the complexes assemble spontaneously into stable 1D supramolecular nanostructures. The ordered accumulation of these supramolecular nanostructures forms large needle-like nanocrystals.

Brown cocrystals of the HAuBr₄·2($i$Pr₂O)⊂2β-CD adduct were also obtained after layering $i$Pr₂O on top of an aqueous solution of the [AuBr₄]⁻⊂β-CD complex for 12 h. X-Ray crystallography reveals that the HAuBr₄·2($i$Pr₂O)⊂2β-CD complex adopts (Supplementary Table 5) the same triclinic space group *P*1 as that of the HAuBr₄·DBC⊂2β-CD complex. In its solid-state superstructure, the [AuBr₄]⁻ anions are located (Fig. 4b) between the primary faces of two β-CD tori, sustained by the multiple [C−H···Br−Au] interactions, an observation which is similar to that in the crystal superstructure of the HAuBr₄·DBC⊂2β-CD complex. The DFT-calculated binding energy between β-CD and [AuBr₄]⁻ in the HAuBr₄·2($i$Pr₂O)⊂2β-CD cocrystal, however, is lower (Supplementary Fig. 47) than that in the HAuBr₄·DBC⊂2β-CD cocrystal. Two $i$Pr₂O molecules are positioned (Fig. 4b) in the cavities and secondary faces of a β-CD dimer on account of intermolecular [C−H···O] hydrogen bonds and the hydrophobic effect. The binding energy between β-CD and $i$Pr₂O is much lower (Supplementary Fig. 49) than that between β-CD and DBC. With the [AuBr₄]⁻ anions and $i$Pr₂O located alternately at their primary and secondary faces, the β-CD tori are arranged (Supplementary Fig. 5e) repeatedly in the order head-to-head and tail-to-tail along the *c* axis. Notably, the space groups of the HAuBr₄·2($i$Pr₂O)⊂2β-CD cocrystals change (Supplementary Table 5) from triclinic *P*1 to monoclinic *P*2₁ after the cocrystals remain in their mother liquid for 3 days, indicating the crystals undergo (Fig. 4a) a

cocrystal-to-cocrystal transformation. For convenience, the initial and transformed cocrystals are described as cocrystal A (Fig. 4b) and cocrystal B (Fig. 4c). In the solid-state superstructure of cocrystal B, only the [AuBr₄]⁻ anions and β-CD were observed (Fig. 4c), whereas $i$Pr₂O was absent. The [AuBr₄]⁻ anions are located (Fig. 4c and Supplementary Fig. 7) in the interspace between the primary faces of two β-CD tori, while the lattice space between the secondary faces of the two β-CD tori is occupied (Fig. 4c) by disordered H₂O molecules. The disordered H₂O molecules take over (Fig. 4a) the role of $i$Pr₂O, holding the two secondary faces of the β-CD tori together. The loss of $i$Pr₂O may be the reason for triggering the cocrystal transformation. Notably, the occupancy for the [AuBr₄]⁻ anions is (Fig. 4c) 50% in cocrystal B, while the [AuBr₄]⁻ anions achieve (Fig. 4b) full occupancy in cocrystal A, demonstrating half of the [AuBr₄]⁻ anions escape into solution during cocrystal transformation. The reason may be because of the lower binding energy (Supplementary Figs. 47, 48) between the central [AuBr₄]⁻ anions and their surrounding species in cocrystal B compared with cocrystal A.

The PXRD confirms the cocrystal-to-cocrystal transformation. The PXRD patterns of the co-precipitates, obtained by adding $i$Pr₂O to the mixture of β-CD and [AuBr₄]⁻ anions, change (Fig. 4d) over time. The HAuBr₄·2($i$Pr₂O)⊂2β-CD suspension, which settles at the bottom of vials after standing 0.5 h, was subjected to powder XRD analysis. Its PXRD pattern is consistent (Fig. 4d) with the simulated pattern based on the single-crystal X-ray data for HAuBr₄·2($i$Pr₂O)⊂2β-CD (cocrystal A), indicating that the co-precipitate possesses the same

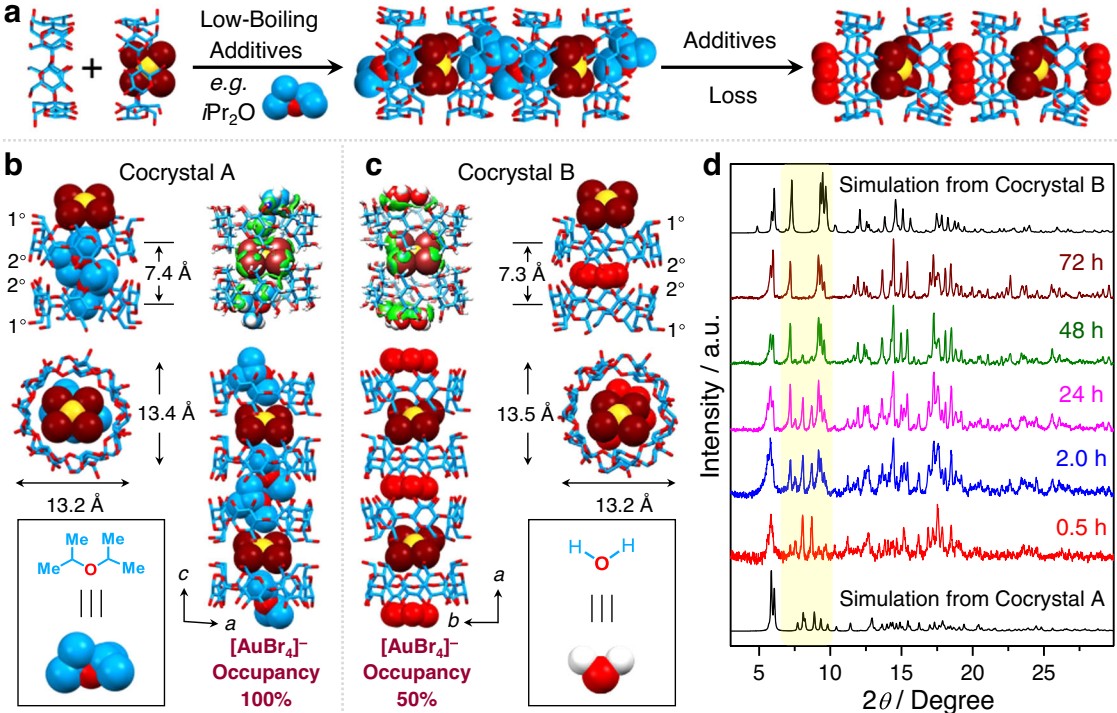

**Fig. 4 | Cocrystal-to-cocrystal transformation of the ternary adducts.** It shows the possible reason for different additives leading to variable gold-recovery efficiencies. **a** Schematic illustration of the cocrystal-to-cocrystal transformation upon loss of additives. **b** Solid-state superstructure and the intermolecular binding iso-surface of the HAuBr₄·2(iPr₂O)⊂2β-CD cocrystal, dubbed as cocrystal A, obtained by adding iPr₂O into the solution of β-CD and [AuBr₄]⁻ followed by standing at room temperature for 12 h. **c** Solid-state superstructure and the intermolecular binding iso-surface of 0.5(HAuBr₄)⊂2β-CD cocrystal, dubbed as cocrystal B,

obtained by adding iPr₂O into the solution of β-CD and [AuBr₄]⁻ followed by standing at room temperature for 3 days. Solvent molecules have been omitted for the sake of clarity. The "1°" represents the primary face and the "2°" represents the secondary face of β-CD. H white, C skyblue, O red, Br brown, Au yellow. **d** Changes in powder X-ray diffraction patterns of the HAuBr₄·2(iPr₂O)⊂2β-CD cocrystal over time, demonstrating that the cocrystal undergoes a phase transformation from cocrystals A to B. The highlighted region with yellow shows the changes in characteristic peaks of cocrystals A and B over time.

superstructure as cocrystal A. When the HAuBr₄·2(iPr₂O)⊂2β-CD suspension was allowed to stand at room temperature for 2.0, 24, and 48 h, the intensity of the feature peaks at 8.1° and 8.7° for cocrystal A became (Fig. 4d) lower and lower, while a set of new peaks appeared gradually at 7.2° and 9.4°. The PXRD pattern of the co-precipitates remained unchanged after standing for 2 days. The final pattern is analogous (Fig. 4d) to the simulation derived from the single-crystal X-ray crystallographic data for cocrystal B. These observations demonstrate that co-precipitates transform (Fig. 4a) spontaneously from cocrystal A to cocrystal B within 2 days. Combined with the solid-state superstructures of cocrystals A and B, we conclude that (i) iPr₂O molecules, bound inside the cavity of β-CD, have been replaced by disordered H₂O molecules during the cocrystal-to-cocrystal transformation, (ii) ~50% of [AuBr₄]⁻ anions escape from the co-precipitates into aqueous solutions during the transformation. These observations may be the reason for the yield of gold-bearing co-precipitate obtained upon adding iPr₂O being lower (Fig. 2b) than that upon adding DBC. According to PXRD patterns, the co-precipitates obtained by adding other low-boiling solvents, e.g., hexane, CH₂Cl₂, and CHCl₃, also undergo (Supplementary Figs. 24–26) the cocrystal-to-cocrystal transformation over time, while the co-precipitates obtained on adding high-boiling solvents, e.g., DBC, benzene and toluene, exhibit (Supplementary Figs. 22, 27 and 28) no obvious changes over time. In the X-ray photoelectron spectra of the co-precipitates obtained by adding CH₂Cl₂ (Supplementary Fig. 29) and CHCl₃ (Supplementary Fig. 30), the signal for chlorine is absent, confirming the loss of CH₂Cl₂ and CHCl₃ during the cocrystal transformation. Additionally, the yields of gold-bearing co-precipitates, after adding high-boiling solvents, are much higher (Fig. 2b) than those upon adding low-boiling solvents,

suggesting that the cocrystal-to-cocrystal transformation is unfavorable to gold recovery.

We believe that several factors need to be considered when choosing suitable additives to recover gold based on our results. They include—(i) Additives that should be hydrophobic and possess a relatively high binding affinity for β-CD, which is a prerequisite for them to participate in the co-assembly process. (ii) Additives should be size-matched with the cavity of β-CD and able to share the cavity together with the [AuBr₄]⁻ anion, which is essential for the formation of 1D supramolecular nanostructures. (iii) Additives with high boiling points are preferred when it comes to improving the stability of the cocrystals. (iv) Additives should be cheap and readily available, so as to reduce costs. (v) Additives should be eco-friendly, an attribute that is vital to sustainable development and environmental protection.

## Gold recovery from gold-bearing scrap
In order to develop a feasible gold-recovery protocol based on additive-induced supramolecular polymerization, the influence of CD and gold salts on gold-recovery efficiency was also investigated. The β-CD-to-KAuBr₄ ratio was optimized first of all. Six samples with different molar ratios of β-CD-to-KAuBr₄ from 0.5 to 3 were prepared. After adding 0.1% (v/v) of DBC, all six clear solutions turned cloudy. The UV–Vis absorption spectra reveal (Fig. 5b) that the gold-recovery efficiencies, based on co-precipitates, increase gradually from 23.8% to 99.0% upon changing the molar ratio of β-CD-to-KAuBr₄ from 0.5 to 3.0. Notably, the gold-recovery efficiency reaches a plateau when the molar ratios rise to (Fig. 5b) 2.5, suggesting that two β-CD tori are required to complex with each [AuBr₄]⁻ anion. γ-CD with a larger binding cavity and a lower binding affinity ($1.39 \times 10^3$ M⁻¹,

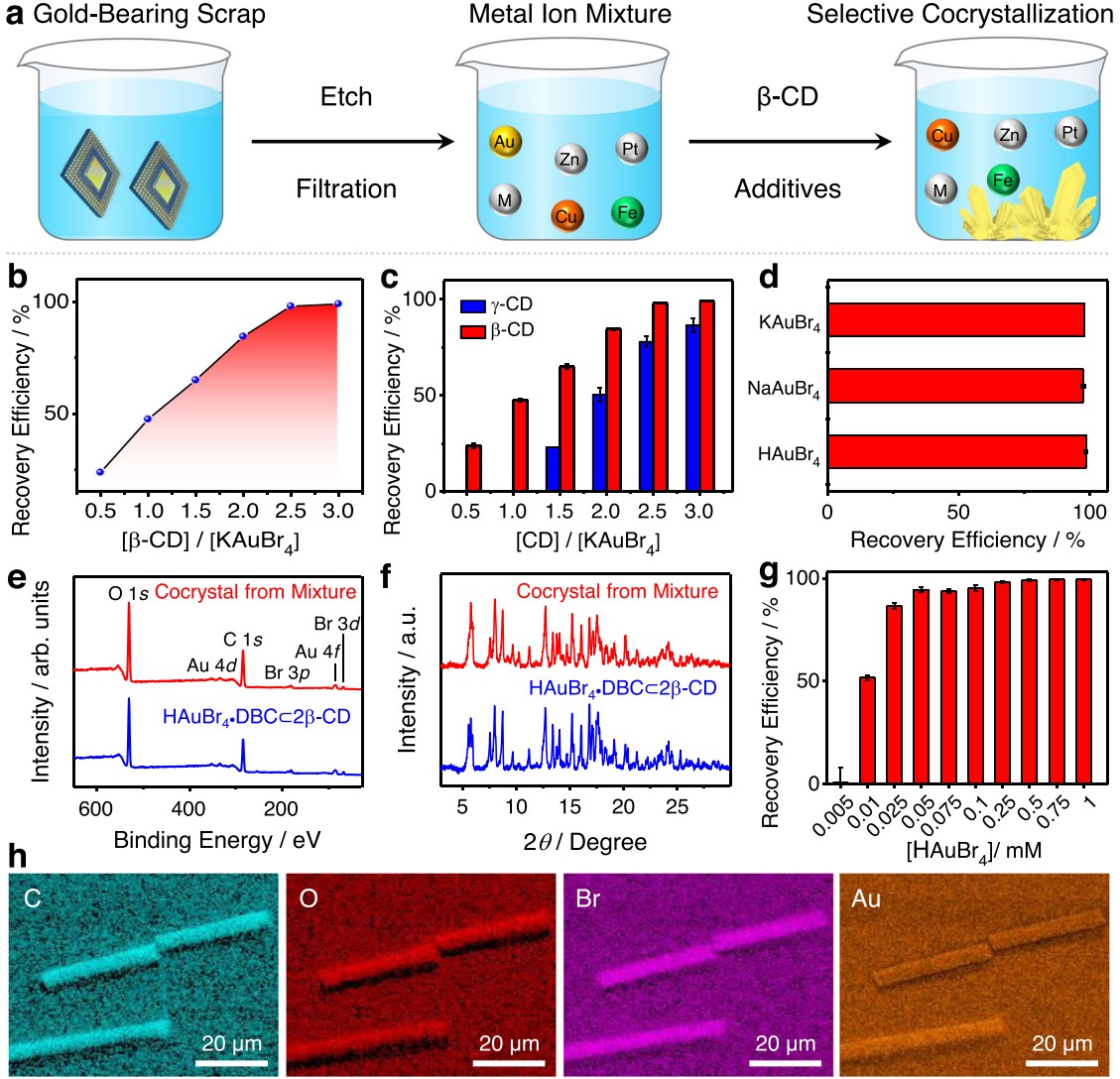

**Fig. 5 | Gold recovery from electronic scrap based on additive-induced supra-molecular polymerization. a** Schematic diagram of the selective recovery of gold from gold-bearing electronic waste using β-CD and additives. **b** Effect of the β-CD-to-[AuBr₄]⁻ ratio on the gold-recovery efficiency. **c** Effect of the CD's size on the gold-recovery efficiency at different molar ratios. The error bar represents the standard deviation of two independent measurements. **d** Effect of changing the counter cation associated with the [AuBr₄]⁻ anion on the gold-recovery efficiency. The error bar represents the standard deviation of two independent measurements. **e** XPS Spectra of HAuBr₄·DBC⊂2β-CD cocrystals (blue), and the microcrystals (red) were obtained by adding β-CD and DBC to a gold-bearing mixture solution. **f** Powder X-ray diffraction patterns of HAuBr₄·DBC⊂2β-CD cocrystals (blue) and the microcrystals (red) were obtained by adding β-CD and DBC to a gold-bearing mixture solution. **g** Effect of changes in concentration of [AuBr₄]⁻ anions on gold-recovery efficiency, when adding β-CD and DBC to a gold-bearing mixture solution. The error bar represents the standard deviation of two independent measurements. **h** SEM-EDS Elemental maps of the HAuBr₄·DBC⊂2β-CD microcrystals obtained by adding β-CD and DBC to a gold-bearing mixture solution.

Supplementary Figs. 16, 17) for [AuBr₄]⁻ anion was also investigated. The gold-recovery efficiencies based on γ-CD are much lower (Fig. 5c) than those obtained when using β-CD at the same CD-to-KAuBr₄ molar ratio, indicating that β-CD is a better candidate for gold recovery than γ-CD.

The effect of the countercation of the gold salts on gold-recovery efficiency has also been investigated. After adding 0.1% (v/v) of DBC to the two aqueous solutions containing NaAuBr₄⊂β-CD and HAuBr₄⊂β-CD complexes, a large amount of brown co-precipitates was formed in both solutions. The corresponding gold-recovery efficiencies (Fig. 5d) are 97.5% (NaAuBr₄) and 98.5% (HAuBr₄), values which are almost identical to that (98.0%) for KAuBr₄. This observation suggests that the additive-induced supramolecular polymerization is independent of the cation associated with the gold salts. Considering that the direct recovery of gold from the leaching solution at low concentration could be a challenge to the industry, we investigated the effect of the

concentration of [AuBr₄]⁻ anions on gold-recovery efficiency. UV–Vis absorption spectra reveal (Supplementary Fig. 21) that over 91.5% of gold in the solutions can be precipitated when the concentration of KAuBr₄ is ≥0.05 mM (9.3 ppm). This observation shows that the technology can be used to recover gold from leaching solutions at low concentrations. Compared with α-CD-based gold-recovery technology[54], which requires high concentrations of KAuBr₄ (>6 mM), the working concentration of [AuBr₄]⁻ anions in the current β-CD-based gold-recovery technology is reduced by a factor of 120.

The high performance of additive-induced cocrystallization between β-CD and [AuBr₄]⁻ motivated us to test the practicability of recovering gold from electronic scrap. A spent gold-bearing alloy cable was employed as a sample for developing a laboratory-scale gold-recovery protocol. The cable was etched (Fig. 5a) using a mixture solution of HBr and H₂O₂ to convert Au into HAuBr₄[56]. The acid concentration of the HAuBr₄-containing solution was adjusted to 1 M.

Insoluble impurities were removed by filtration. After adding an aqueous solution of β-CD and 0.1% (v/v) of DBC to the gold-bearing filtrate, yellow microcrystals formed (Fig. 5a) immediately.

In the X-ray photoelectron spectrum of the microcrystals, only the characteristic peaks of carbon, oxygen, bromine, and gold were observed (Fig. 5e), while signals for other metals were absent. The FTIR spectrum of the microcrystals matches (Supplementary Fig. 32) well with that of the HAuBr$_4$·DBC⊂2β-CD cocrystal, indicating they have similar components. The similar PXRD patterns (Fig. 5f) between the microcrystals and the HAuBr$_4$·DBC⊂2β-CD cocrystal demonstrate that the microcrystals obtained from the mixture possess the same solid-state superstructure as the HAuBr$_4$·DBC⊂2β-CD cocrystal. These observations indicate that the additive-induced supramolecular polymerization between HAuBr$_4$, DBC, and β-CD is highly selective and that the presence of large amounts of other metals has a negligible impact on the additive-induced crystallization process. SEM Analysis uncovers that the microstructure of the yellow microcrystals is similar to that of the HAuBr$_4$·DBC⊂2β-CD adduct. Many microrods were observed (Supplementary Fig. 35) in the SEM images. SEM-EDS Revealed (Fig. 5h) that all the elements, including carbon, oxygen, bromine, and gold, were distributed uniformly in the microrods. The microscopic investigation and elemental analysis confirmed the high specificity of the co-crystallization. The ICP-MS analysis (Fig. 5g) indicated 99.0% of the [AuBr$_4$]$^-$ anions are separated from the leaching solution at a concentration of 0.5 mM (93 ppm), a value which is close to the precipitation efficiency (99.8%) obtained when adding 0.1% (v/v) of DBC to a solution of the KAuBr$_4$⊂β-CD complex. In order to demonstrate the applicability of the additive-induced supramolecular polymerization to recover a smaller amount of gold, two mixtures containing 9.3 and 4.7 ppm of gold were prepared. ICP-MS Analysis indicates (Fig. 5g) that 94.4% and 86.6% of the gold in them were precipitated. These results suggest that additive-induced polymerization can be used to recover the gold from low-grade gold ores and low-concentration gold-bearing electronic scrap. Finally, the gold metal was recovered from the yellow co-precipitate after reduction with N$_2$H$_4$·H$_2$O. The β-CD can be recycled by precipitating with acetone and followed by recrystallization. Based on this laboratory-scale gold-recovery experiment, a gold-recovery flow diagram (Supplementary Fig. 52) has been proposed. The flow diagram provides us with a reference to develop a feasible protocol for recovering gold on a larger scale.

## Discussion

An eco-friendly and sustainable supramolecular metallurgical technology for gold recovery has been demonstrated. It is based on an additive-induced supramolecular polymerization of second-sphere coordinated adducts formed between β-cyclodextrin and tetrabromoaurate anions. The additives drive the assembly by obliging the tetrabromoaurate anions to move from the inner cavity to the primary face of two β-cyclodextrin tori, while themselves occupying the space between the secondary faces of two β-cyclodextrin tori, leading to the formation of infinite 1D supramolecular nanostructures that precipitate from the aqueous solutions as cocrystals. This additive-induced supramolecular polymerization, not only provides a feasible method for the rapid crystallization of the supramolecular polymers but also opens the door for regulating the assembly behavior of second-sphere coordinated adducts.

In contrast to the traditional antisolvent precipitation method, the additive-induced polymerization reported in this research provides the following advantages—(i) It only requires a small volume fraction (<0.3%) of additives to effectively precipitate the target compound with a high recovery efficiency (>99.5%). (ii) The additives do not have to be miscible with the solution. (iii) The molecular recognition-driven supramolecular polymerization is highly selective for the precipitation of target compounds in the presence of other structurally similar substrates.

From a practical viewpoint, this research describes a highly efficient and sustainable gold-recovery protocol. Compared with our previously reported[54] protocol using α-cyclodextrin, the additive-induced polymerization with β-cyclodextrin comes with the following attributes—(i) The gold-recovery can be performed at a low concentration (9.3 ppm) with much better recovery efficiency (>94%). (ii) No additional potassium ions are needed. (iii) Co-precipitation can be performed directly in acidic leaching solutions without the need for neutralization. (iv) The cost of β-cyclodextrin is lower than that of α-cyclodextrin. In summary, our establishment of additive-induced polymerization constitutes an attractive strategy for the practical recovery of gold and leads to significantly reduced energy consumption, cost inputs, and environmental pollution. We are currently optimizing the strategy to recover gold from lower-concentration gold-bearing e-waste and exploring the generality of this strategy to separate other target metal ions.

## Methods

### Materials

The compounds HAuBr$_4$ (99.9%, Alfa Aesar), NaAuBr$_4$ (99.9%, Strem Chemicals), KAuBr$_4$ (99.9%, Sigma), K$_2$PdBr$_4$ (98.0%, Strem Chemicals), K$_2$PtBr$_4$ (99.0%, Strem Chemicals), β-CD (>98.0%, Oakwood Chemical), γ-CD (>98.0%, Sigma), DBr (47 wt%, Sigma) in D$_2$O, HBr (48 wt%, Sigma) in H$_2$O, and H$_2$O$_2$ (30 wt%, Oakwood Chemical) in H$_2$O were purchased from commercial suppliers and used without further purification. All the additives, i.e., dibutyl carbitol (DBC), isopropyl ether (iPr$_2$O), diethyl ether (Et$_2$O), hexane, ethyl acetate, dichloromethane (CH$_2$Cl$_2$), chloroform (CHCl$_3$), benzene, toluene, olive oil, vegetable oil, and pump oil are commercially available. Ultra-pure water was generated by a Milli-Q system.

### UV−Vis absorption spectroscopy

UV−Vis absorption spectra were recorded in 1 M HBr aqueous solutions at 298 K. UV−Vis absorption spectra were recorded on a UV-3600 Shimadzu spectrophotometer in rectangular quartz cells with light paths of 4 mm. Each gold-recovery experiment was duplicated independently, and the average gold-recovery efficiencies are presented with their standard deviations.

### Fourier-transform infrared spectroscopy

Fourier-transform infrared (FT-IR) spectroscopy was performed on a Nexus 870 spectrometer (Thermo Nicolet) in the mode of attenuated total reflection (ATR) with the range from 4000 to 600 cm$^{-1}$ and at a resolution of 0.125 cm$^{-1}$.

### High-resolution mass spectrometry

High-resolution mass spectrum (HRMS) for the host−guest complex was recorded on an Agilent 6210 Time-of-Flight (TOF) LC-MS with an ESI source.

### NMR spectroscopy

NMR spectra were recorded on a Bruker Avance III 600 MHz spectrometer in D$_2$O containing 0.5 M DBr. Chemicals shifts ($\delta$) are given in ppm with residual H$_2$O signals as a reference. $^1$H NMR Titrations: Highly-concentrated solutions of KAuBr$_4$, K$_2$PdBr$_4$, or K$_2$PtBr$_4$ in D$_2$O (containing 0.5 M DBr) as the titrating solution were added dropwise to a D$_2$O solution (containing 0.5 M DBr) of β-CD or γ-CD. Binding constants were obtained by fitting a 1:1 isotherm according to the programs available at http://app.supramolecular.org/bindfit/.

### Crystallization and single-crystal X-ray diffraction analyses

The KAuBr4⊂β-CD complex. Brown single crystals were obtained by slowly cooling an aqueous solution of KAuBr$_4$ and β-CD from 90 °C to room temperature over 6 h. The HAuBr$_4$·DBC⊂2β-CD complex: Brown single crystals were obtained by slow vapor diffusion of DBC into an

aqueous solution of $KAuBr_4$ containing 2 molar equivalents of β-CD over 3 days. The $HAuBr_4$·2($iPr_2O$)⊂2β-CD complex (cocrystal A): Brown single crystals were obtained by slow liquid-liquid diffusion of $iPr_2O$ into an aqueous solution containing $KAuBr_4$ and 2 molar equivalents of β-CD over 12 h. The 0.5($HAuBr_4$)⊂2β-CD complex (cocrystal B): Brown single crystals were obtained by allowing the $HAuBr_4$·2($iPr_2O$)⊂2β-CD suspension to stand at room temperature for 3 days. The suitable crystals were mounted on a MITIGEN holder in Paratone oil on a Rigaku XtaLAB Synergy, Single source at home/near, HyPix diffractometer using CuKα ($\lambda = 1.5418$ Å) or MoKα ($\lambda = 0.7107$ Å) radiation. Data were collected using the Bruker APEX-II or Rigaku CrysAlis Pro program. The superstructures were solved with the ShelXT program using intrinsic phasing and refined with the ShelXL refinement package using least-squares minimization in OLEX2 software.

## Powder X-ray diffraction analysis
Powder X-ray diffraction (PXRD) analyses were performed on an STOE-STADI MP powder diffractometer equipped with an asymmetrically curved germanium monochromator (Cu-Kα1 radiation, $\lambda = 1.54056$ Å) and a one-dimensional silicon strip detector (MYTHEN2 1K from DECTRIS). Samples for superstructural analysis were measured at room temperature in transmission mode. The simulated PXRD patterns were calculated using Mercury software 4.3.0.

## X-ray photoelectron spectroscopy
X-ray photoelectron spectroscopic (XPS) analyses were conducted on a fully digital state-of-the-art X-ray photoelectron spectrometer (Thermo Scientific ESCALAB 250Xi), which was equipped with an electron flood gun and a scanning ion gun. The diameter of the X-ray spot was 500 μm, and the scan range was from 0 to 1200 eV.

## Thermogravimetric analysis
Thermogravimetric analysis (TGA) experiments were performed on a Mettler Toledo TGA/DSC1 Stare System (Schwerzenbach, Switzerland) interfaced with a PC using Stare software. Samples were placed in an $Al_2O_3$ crucible and heated at a rate of 15 K min$^{-1}$ from 40 to 800 °C under a helium atmosphere.

## Inductively coupled plasma mass spectrometry
ICP-MS was performed on a computer-controlled (QTEGRA software) Thermo iCapQ ICP-MS (Thermo Fisher Scientific, Waltham, MA, USA) operating in STD mode and equipped with an ESI SC-2DX PrepFAST autosampler (Omaha, NE, USA). The internal standard was added inline using the prepFAST system and consisted of 1 ng/mL of a mixed-element solution containing Bi, In, $^6$Li, Sc, Tb, Y (IV-ICPMS-71D from Inorganic Ventures). Online dilution was also carried out by the prepFAST system and used to generate a calibration curve consisting of 100, 50, 20, 10, 5, and 1 ppb Au. Each sample was acquired using one survey run (10 sweeps) and three main (peak jumping) runs (40 sweeps). The isotopes selected for analysis were $^{197}$Au, $^{89}$Y, $^{115}$In, $^{159}$Tb, and $^{209}$Bi (chosen as internal standards for data interpolation and machine stability). Instrument performance was optimized daily by auto-tuning, followed by verification with the aid of a performance report.

## Scanning electron microscopy
Scanning electron microscopic (SEM) images were obtained on a SU8030 scanning electron microscope at the voltages of 10/15 kV, while the energy-dispersive X-ray spectroscopy (EDS) elemental maps were recorded at 15 kV.

## Density functional theory calculations
The superstructures from the single-crystal X-ray diffraction were used for the density functional theory (DFT) calculations in the Orca program (version 4.1.2) using the hybrid generalized gradient approximation (GGA) Becke three-parameter Lee-Yang-Parr (B3LYP) functional, the Ahlrichs double-zeta basis set with a polarization function Def2-SVP, and Grimme's third-generation atom-pairwise dispersion correction with Becke Johnson damping (D3BJ); an integration grid of four was used throughout. To further speed up the DFT optimizations, the Coulomb integral and numerical chain-of-sphere integration for the HF exchange (RIJCOSX) method was applied with the Def2/J auxiliary basis.

## Additive-induced ternary cocrystallization experiments
Aqueous stock solutions of $KAuBr_4$ were prepared by dissolving directly the corresponding commercially available salt in an aqueous HBr (2 M) solution, while an aqueous solution of β-CD was prepared by dissolving the β-CD powder in ultra-pure water. The addition of 2 molar equivalents of β-CD (1 mL, 10 mM) to a $KAuBr_4$ (1 mL, 5 mM) aqueous solution led to the formation of a 1:1 $KAuBr_4$⊂β-CD complex. When specific additives (0.1% v/v) were added to the resulting aqueous solutions, yellow suspensions were formed immediately. The yellow solids were isolated by filtration, washed, and air-dried. The concentrations of [$AuBr_4$]$^-$ remaining in the filtrates were determined by UV–Vis absorption spectroscopy or ICP-MS analysis. The metal precipitation yields were calculated based on the aqueous solution's initial and residual concentrations of metal anions.

## Gold recovery from scrap
A gold-bearing cable was obtained from the local electronic junk shop. The yellow cable (15 mg) was first leached with a mixture of HBr and $H_2O_2$ overnight. Subsequently, the acid concentration in the $HAuBr_4$-containing leaching solution was adjusted to 1 M with ultra-pure $H_2O$, and the insoluble impurities were removed by filtration. A saturated aqueous solution of β-CD containing ~1 M HBr was added to the leaching solution, forming the $HAuBr_4$⊂β-CD complex. Upon adding 0.1% (v/v) of DBC, the solution became gradually cloudy. After stirring for 5 min, the co-precipitate of $HAuBr_4$·DBC⊂2β-CD was separated from other metals by filtration and washed with ultra-pure $H_2O$. The metals trapped in the co-precipitates were analyzed by ICP-MS by comparing them with the metal concentrations of the solution before and after adding DBC. In order to convert the [$AuBr_4$]$^-$ anions trapped in the co-precipitate to gold metal, the $HAuBr_4$·DBC⊂2β-CD co-precipitates were dispersed in an aqueous solution and reduced with $N_2H_4$·$H_2O$. After centrifugation and washing with $H_2O$ to dissolve the residual β-CD, gold metal was obtained. β-CD, which dissolves in the aqueous solution, can be recycled by precipitating with acetone and followed by recrystallization.

# Data availability
All the data supporting the conclusions are included in this article and its Supplementary files. Raw data are available from corresponding authors upon request. Crystallographic data for the structures reported in this Article have been deposited at the Cambridge Crystallographic Data Centre, under deposition numbers CCDC 2206843 ($KAuBr_4$⊂β-CD), 2206844 ($HAuBr_4$·DBC⊂2β-CD), 2206845 ($HAuBr_4$·2($iPr_2O$)⊂2β-CD), and 2206846 (0.5($HAuBr_4$)⊂2β-CD). Copies of the data can be obtained free of charge via https://www.ccdc.cam.ac.uk/structures/. Checkcif files (Supplementary Data 1–4) for the four crystals are attached. A real-time video (Supplementary Movie 1) recording of the formation of co-precipitates between $KAuBr_4$ and β-CD when adding DBC is attached in the form of an individual Supplementary MP4 file.

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

## Acknowledgements

The authors thank Northwestern University (NU) for their support of this research, which made use of the IMSERC X-Ray facility at NU. It receives backing from the Soft and Hybrid Nanotechnology Experimental (SHyNE) Resource (NSF ECCS-1542205) and NU. H.W., L.O.J., W.L., G.C.S., and J.F.S. also acknowledge the strong support provided by the Center for Sustainable Separations of Metals (CSSM), a National Science Foundation (NSF) Center for Chemical Innovation (CCI), grant number CHE-1925708. Theoretical investigations were supported in part by the computational resources and staff contributions provided by the NU Quest High-Performance Computing Facility, which is jointly supported by the Office of the Provost, the Office for Research, and Northwestern University Information Technology.

## Author contributions

H.W., Yu.W., and W.L. conceived the research and carried out the major experiments and analyses. Yu.W., L.O.J., and G.C.S. conducted theoretical calculations. C.L.S., and C.T. carried out the X-ray crystallographic analyses. H.W., Yu.W., and W.L. wrote the manuscript. B.S., X.-Y.C., L.Z., and Yo.W. were involved in the discussions and contributed to the manuscript preparation. J.F.S. directed and supervised the research. All authors discussed the experimental results and contributed to the preparation of the manuscript.

## Competing interests

The authors declare no competing interests.
