## [Peer Review File · Nature Communications]

Reviewer comments, first round review –

Reviewer #1 (Remarks to the Author):

The paper by Huang Wu et al report excellent results on the recovery of gold by exploiting the formation of second sphere coordinate adducts obtained by using beta cyclodextrin and tetrabromoaurate anion in the presence of selected solvent.

The article can have a great and positive impact on the recovery of gold by using a more eco friendly technology. The manuscript is well organized and written. The experimental data are impressive and the considerations completely supported by the data. The reference list adequate. For sure the paper deserves for a publication on Nature Communications however I would like to ask few minor revision to the authors. More in details:

- 1) What is the thermal stability of the adduct formed?
- 2) The cocrystallization is very selective towards AuBr_4^- anion, this means that similar anions like PdBr_4^- and PtBr_4^- have a lower inclusion constant?
- 3) The authors report the characteristics of the solvent that allow better coprecipitation. It is possible to imagine the influence of LogP?
- 4) What happens if higher amount of DBC is used? Cocrystallization is prevented?
- 5) γCD works worse than βCD . This is a good point because γCD is much more expensive than βCD . However, it depends from lower stability constant with AuBr_4^- anion?

Reviewer #2 (Remarks to the Author):

The authors report a gold recovery protocol based on the gold precipitation of a second sphere coordinated adduct formed between β -cyclodextrin and tetrabromoaurate anions in the presence of appropriate additives. In particular, when dibutyl carbitol (DBT) is used, a 99.8% capture efficiency is achieved. Overall, the results are highly interesting and the manuscript is well-written. It is true that this work reminded me a previous work of some of the authors (Ref 51) based on the second sphere adduct formation with alpha-cyclodextrin. However, the authors have greatly improved both the protocol and the capture efficiency, which, in my opinion is a key parameter to suggest to accept the manuscript in such a reputed journal as Nature Communications.

Reviewer #3 (Remarks to the Author):

In this manuscript, the authors report a highly selective gold-recovery technology based on the formation of supramolecular adducts between beta-cyclodextrin, tetrabromoaurate anions and an additive. When dibutyl carbitol was used as additive, 99.8 % of gold in the solution can be recovered. This high efficiency was explained by considering an additive-induced supramolecular polymerization of adducts formed between beta-cyclodextrin and tetrabromoaurate. This mechanism was fully supported by NMR, UV-vis, FT-IR, SEM and crystal X-ray diffraction analysis. I highly recommend the publication of this very nice work. Indeed, the approach is very original and the proposed mechanism particularly relevant. I have only two comments: 1) Could Cocrystal B be obtained directly from a solution of beta-cyclodextrin and tetrabromoaurate anions without using low-boiling solvents (and, consequently, without passing through cocrystal A)? 2) How do the authors envisage recovering the gold once the precipitate is recovered? by calcination? This should be indicated in the manuscript.

Reviewer #4 (Remarks to the Author):

The aim of the present study was to systematically examine the effects of variations in the process parameters of the antisolvent precipitation method employed in the high-efficiency gold recovery. The golden triangle "WHAT, WHY, HOW" is well discussed in the introduction, surprisingly, with some essential references being omitted (vide infra). The presented work is a valuable extension

of the previous contributions to this topic. For example, Stoddart and coworkers reported an alkali metal cation-dependent approach to gold recovery, facilitated by second-sphere coordination with α -cyclodextrin (J. Am. Chem. Soc. 2016, 138, 11643; I wonder why this work is not referred in the submitted manuscript). In this work the authors noted that "when aqueous solutions of each of KAuCl_4 and KAuBr_4 as well as aqueous solutions of α -, β -, and γ -CDs in turn are mixed, we observed that, after only a few minutes, the mixture of KAuBr_4 and α -CD leads to a pale brown co-precipitate, while all the other five mixtures remain as clear solutions in water."

In the present manuscript, the authors are completing previous studies and demonstrate that $[\text{AuBr}_4]^-$ anion is efficiently encapsulated by β -CD in an aqueous solutions. Moreover, brown single crystals of a 1:1 AuBr_4^-/β -CD complex were obtained by slowly cooling an aqueous solution and characterized using single crystal X-ray diffraction analysis. Then they revealed that upon adding a small amount of common organic solvents to an aqueous solution of the KAuBr_4/β -CD complex, brown co-precipitates form immediately. Thus, for accelerating dissolution of this kind of the host-guest complex, an antisolvent precipitation method was applied. In order to verify the generality of this antisolvent-induced co-precipitation behavior, various organic solvents were tested. Gold-recovery efficiencies ranging from 27.0 to 99.8 %, depending on the antisolvents. When a small amount of diethylene glycol dibutyl ether (DBC) was used as the antisolvent, the $[\text{AuBr}_4]^-$ anions were almost entirely precipitated from aqueous solutions with a gold-recovery efficiency of 99.8%. The effect of the counteraction of the gold salts on gold-recovery efficiency was also investigated. Finally, the mechanism for the DBC-induced supramolecular polymerization was elucidated by X-ray crystallography and this study revealed an intriguing role of antisolvent molecules on the assembly pathway. Based on this analysis, the authors named the applied antisolvent method as an additive-induced supramolecular polymerization.

In conclusion, the reported investigations are well planned, crafted, and meticulously executed. While I do not question the validity of this work, its scientific novelty is rather limited, and I'm afraid I am not persuaded that the reported findings represent a sufficiently striking advance to justify publication in Nat. Commun.

Some additional remarks:

The authors refer in the introduction part to a few review articles on self-assemblies based on host-guest molecular recognition motifs, however they surprisingly omit those relating to CDs-based systems, e.g., Chem. Rev. 2015, 115, 15, 7240; <https://doi.org/10.1021/cr5005315>, and Chem. Rev. 2017, 117, 22, 13461; <https://doi.org/10.1021/acs.chemrev.7b00231>

Response Letter to Reviewers' Comments

Reply to Reviewer #1

Comments: The paper by Huang Wu et al report excellent results on the recovery of gold by exploiting the formation of second sphere coordinate adducts obtained by using beta cyclodextrin and tetrabromoaurate anion in the presence of selected solvent. The article can have a great and positive impact on the recovery of gold by using a more eco-friendly technology. The manuscript is well organized and written. The experimental data are impressive and the considerations are completely supported by the data. The reference list is adequate. For sure the paper deserves for a publication on Nature Communications however I would like to ask for a few minor revisions to the authors. More in details:

Response: We thank the reviewer for referring to our research in a positive manner and providing valuable comments. Following the reviewer's comments, we have revised the manuscript carefully and added some new experimental data and discussions.

Comment 1: What is the thermal stability of the adduct formed?

Response: We have performed thermogravimetric analysis (TGA) in order to investigate the thermal stabilities of the co-precipitates upon adding different additives. The TGA profile for the $\text{KAuBr}_4 \cdot \text{DBC} \cdot 2\beta\text{-CD}$ co-precipitate reveals (Supplementary Fig. 51a) that it begins to suffer mass loss at temperatures around 100 °C, most likely because of the loss of crystalline water. Significant decomposition occurs around 160 and 280 °C, arising from halide release and the breakdown of $\beta\text{-CD}$. Finally, over 70 wt% of the original mass of the co-precipitate was lost at 800 °C. The TGA traces for the co-precipitates obtained by adding other additives behave in a similar fashion (Supplementary Fig. 51b) to that of the $\text{KAuBr}_4 \cdot \text{DBC} \cdot 2\beta\text{-CD}$ adduct over the temperature range from 40 to 800 °C, indicating they have similar components. We have added appropriate statements on Page 7 of the revised Main Text and Supplementary Fig. 51 in the revised Supporting Information.

Supplementary Figure 51 | Thermogravimetric analysis (TGA) of (a) $\text{KAuBr}_4 \cdot \text{DBC} \cdot 2\beta\text{-CD}$ and (b) the co-precipitates obtained by adding different additives to the aqueous solutions of $\beta\text{-CD}$ and $[\text{AuBr}_4]^-$ anion

Comment 2: The cocrystallization is very selective towards $[\text{AuBr}_4]^-$ anion, this means that similar anions like $[\text{PdBr}_4]^{2-}$ and $[\text{PtBr}_4]^{2-}$ have a lower inclusion constant?

Response: We have carried out ^1H NMR titrations to measure the binding affinities between $\beta\text{-CD}$ and $[\text{PdBr}_4]^{2-}$ / $[\text{PtBr}_4]^{2-}$ anions, respectively. The binding affinities between $\beta\text{-CD}$ and $[\text{PdBr}_4]^{2-}$ / $[\text{PtBr}_4]^{2-}$ anions were determined to be 1.45×10^2 and 33.3 M^{-1} , respectively. These binding affinities are lower than that ($4.47 \times 10^4 \text{ M}^{-1}$) between $\beta\text{-CD}$ and $[\text{AuBr}_4]^-$ anion. These lower binding affinities may be the reason that no precipitate was formed upon adding additives to aqueous solutions of $\beta\text{-CD}$ containing the $[\text{PdBr}_4]^{2-}$ / $[\text{PtBr}_4]^{2-}$ anions. We have added some comments on Page 5 of the revised Main Text and in Supplementary Figs. 12–15 in the revised Supporting Information.

Supplementary Figure 12 | ^1H NMR Spectra (600 MHz, D_2O containing 0.5 M DBr, 25 °C) of $\beta\text{-CD}$ (0.5 mM) titrated with K_2PdBr_4 (100 mM)

Supplementary Figure 13 | (a) Titration isotherm created by monitoring changes in the chemical shift of H-5 in β -CD, caused by the stepwise addition of K_2PdBr_4 at 25 °C. Red line is the result of curve fitting using a 1:1 receptor-substrate binding model. (b) Mole fractions are based on the fitting results, indicating that the concentration of the free β -CD undergoes a continuous decrease (blue trace), while the concentration of $\text{K}_2\text{PdBr}_4 \subset \beta$ -CD complex undergoes a continuous increase (red trace).

Supplementary Figure 14 | ^1H NMR Spectra (600 MHz, D_2O containing 0.5 M DBr, 25 °C) of β -CD (0.15 mM) titrated with a saturated K_2PtBr_4 (11.34 mM) aqueous solution. In order to avoid the error caused by volume dilution, we did not continue to increase the number of equivalents of the K_2PtBr_4 during the titration.

Supplementary Figure 15 | (a) Titration isotherm created by monitoring changes in the chemical shift of H-4 in β -CD, caused by the stepwise addition of K_2PtBr_4 at 25 °C. Red line is the result of curve fitting using a 1:1 receptor-substrate binding model. (b) Mole fractions are based on the fitting results, indicating that the concentration of the free β -CD undergoes a continuous decrease (blue trace), while the concentration of $\text{K}_2\text{PtBr}_4\text{-}\beta\text{-CD}$ complex undergoes a continuous increase (red trace).

Comment 3: The authors report the characteristics of the solvent that allow better co-precipitation. It is possible to imagine the influence of LogP?

Response: We have checked the LogP values for the solvents we used for inducing the formation of co-precipitates. According to the values of LogP, the seven solvents are arranged in the order of hexane (3.76) > toluene (2.68) > benzene (2.13) > chloroform (1.97) > dibutyl carbitol (1.92) > isopropyl ether (1.52) > dichloromethane (1.25). It has not been possible to establish a correlation with the following sequence (Fig. 2b) of the gold-recovery efficiencies of dibutyl carbitol (99.8%) > toluene (96.0%) > benzene (87.0%) > chloroform (83.0%) > dichloromethane (73.5%) > isopropyl ether (68.0%) > hexane (23.0%). We find it difficult to involve the influence of LogP values of solvents on gold recovery at this stage. We have summarized several factors on Page 12 in the revised Main Text that need to be considered when choosing suitable additives to recover gold based on our results.

Comment 4: What happen if higher amount of DBC is used? Cocrystallization is prevented?

Response: We have investigated (Fig. 2c) the effect of increasing the amount of the DBC on the crystallization of the $\text{KAuBr}_4 \cdot \text{DBC} \subset 2\beta\text{-CD}$. After adding 0.05, 0.1, 0.25, 0.5, 1, and 2% (v/v) of DBC to the aqueous solutions of $[\text{AuBr}_4]^- \subset \beta\text{-CD}$ complex, all six clear solutions turned cloudy. The yield of co-precipitate increases gradually from 99.4 to 99.8% upon changing the amounts of DBC from 0.05 to 2%. This observation suggests that higher amounts of DBC do not prevent the additive-induced cocrystallization of $\text{KAuBr}_4 \cdot \text{DBC} \subset 2\beta\text{-CD}$ adduct in our experimental amounts. It is worth mentioning that the amounts of additives should be as little as possible for the simple reason that it will reduce costs.

Comment 5: $\gamma\text{-CD}$ works worse than $\beta\text{-CD}$. This is a good point because $\gamma\text{-CD}$ is much more expensive than $\beta\text{-CD}$. However, it depends from lower stability constant with $[\text{AuBr}_4]^-$ anion?

Response: We have carried out ^1H NMR titration to measure the binding affinity between $\gamma\text{-CD}$ and $[\text{AuBr}_4]^-$ anion. The binding affinity between $\gamma\text{-CD}$ and $[\text{AuBr}_4]^-$ was found to be $1.39 \times 10^3 \text{ M}^{-1}$. The fact that the binding affinity is lower than that ($4.47 \times 10^4 \text{ M}^{-1}$) between $\beta\text{-CD}$ and $[\text{AuBr}_4]^-$ anion may be the reason for the gold-recovery efficiency on using $\gamma\text{-CD}$ is lower than that when employing $\beta\text{-CD}$. We have added some comments on this situation on Page 12 in the revised Main Text and Supplementary Figs. 16–17 in the revised Supporting Information.

Supplementary Figure 16 | ^1H NMR Spectra (600 MHz, D_2O containing 0.5 M DBr , 25 °C) of $\gamma\text{-CD}$ (0.25 mM) titrated with KAuBr_4 (100 mM)

Supplementary Figure 17 | (a) Titration isotherm created by monitoring changes in the chemical shift of H-3 in γ -CD, caused by the stepwise addition of KAuBr_4 at 25 °C. Red line is the result of curve fitting using a 1:1 receptor-substrate binding model. (b) Mole fractions are based on the fitting results, indicating that the concentration of the free γ -CD undergoes a continuous decrease (blue trace), while the concentration of $\text{KAuBr}_4\text{-}\gamma$ -CD complex undergoes a continuous increase (red trace).

Reply to Reviewer #2

Comments: The authors report a gold recovery protocol based on the gold precipitation of a second sphere coordinated adduct formed between β -cyclodextrin and tetrabromoaurate anions in the presence of appropriate additives. In particular, when dibutyl carbitol (DBC) is used, a 99.8% capture efficiency is achieved. Overall, the results are highly interesting and the manuscript is well-written. It is true that this work reminded me a previous work of some of the authors (Ref 51) based on the second sphere adduct formation with alpha-cyclodextrin. However, the authors have greatly improved both the protocol and the capture efficiency, which, in my opinion is a key parameter to suggest to accept the manuscript in such a reputed journal as Nature Communications.

Response: We really appreciate the reviewer for providing valuable comments and referring to our research in a positive manner.

Reply to Reviewer #3

Comments: In this manuscript, the authors report a highly selective gold-recovery technology based on the formation of supramolecular adducts between beta-cyclodextrin, tetrabromoaurate anions and an additive. When dibutyl carbitol was used as additive, 99.8 % of gold in the solution can be recovered. This high efficiency was explained by considering an additive-induced supramolecular polymerization of adducts formed between beta-cyclodextrin and tetrabromoaurate. This mechanism was fully supported by NMR, UV-vis, FT-IR, SEM and crystal X-ray diffraction analysis. I highly recommend the publication of this very nice work. Indeed, the approach is very original and the proposed mechanism particularly relevant.

Response: We thank the reviewer for referring to our research in a positive manner and providing valuable comments. According to the reviewer's comments, some statements have been added to improve the quality of the manuscript.

Comment 1: Could Cocrystal B be obtained directly from a solution of β -cyclodextrin and tetrabromoaurate anions without using low-boiling solvents (and, consequently, without passing through cocrystal A)?

Response: We have attempted to obtain cocrystal B directly from aqueous solutions of β -CD and $[\text{AuBr}_4]^-$. Without using low-boiling solvents, the mixture of β -CD and $[\text{AuBr}_4]^-$ remains clear in an aqueous solution rather than undergoing crystallization. On slow evaporation of water from an aqueous solution of β -CD and KAuBr_4 , a cocrystal with a superstructure identical to that of the $\text{KAuBr}_4 \cdot \beta\text{-CD}$ complex was obtained. Upon slow diffusion of EtOH or MeOH into the aqueous solution of β -CD and KAuBr_4 , crystals of the $\text{KAuBr}_4 \cdot 2\beta\text{-CD}$ adduct were obtained. They adopt the same solid-state superstructure as the crystal (CCDC No.918414) reported in the literature (*Nat. Commun.* **2013**, *4*, 1855). In this solid-state superstructure, the $[\text{AuBr}_4]^-$ anions are centered between the primary faces of two adjacent β -CD tori with an occupancy of 100%, while K^+ ions link the two secondary faces of two adjacent β -CD tori. This superstructure is also different from that of cocrystal B ($0.5(\text{HAuBr}_4) \cdot 2\beta\text{-CD}$), wherein the occupancy of $[\text{AuBr}_4]^-$ anion is 50%, and K^+ ions are absent. It has proved challenging to obtain cocrystal B without using low-boiling solvents.

Comment 2: How do the authors envisage recovering the gold once the precipitate is recovered? by calcination? This should be indicated in the manuscript.

Response: We have added relevant statements on Page 14 of the revised Main Text, on Page 18 of the revised Method Section, and Supplementary Fig. 52 in the revised Supporting Information. The details are as follows: In order to convert the $[\text{AuBr}_4]^-$ anions trapped in the co-precipitate to gold metal, the $\text{HAuBr}_4 \cdot \text{DBC} \subset 2\beta\text{-CD}$ co-precipitates were dispersed in an aqueous solution and reduced with $\text{N}_2\text{H}_4 \cdot \text{H}_2\text{O}$. After centrifugation and washing with H_2O to dissolve the residual $\beta\text{-CD}$, gold metal was obtained. $\beta\text{-CD}$, which dissolves in the aqueous solution, can be recycled by precipitating with acetone and followed by recrystallization.

Reply to Reviewer #4

Comments: The aim of the present study was to systematically examine the effects of variations in the process parameters of the antisolvent precipitation method employed in the high-efficiency gold recovery. The golden triangle “WHAT, WHY, HOW” is well discussed in the introduction, surprisingly, with some essential references being omitted (vide infra). The presented work is a valuable extension of the previous contributions to this topic. For example, Stoddart and coworkers reported an alkali metal cation-dependent approach to gold recovery, facilitated by second-sphere coordination with α -cyclodextrin (J. Am. Chem. Soc. 2016, 138, 11643; I wonder why this work is not referred in the submitted manuscript). In this work the authors noted that “when aqueous solutions of each of KAuCl_4 and KAuBr_4 as well as aqueous solutions of α -, β -, and γ -CDs in turn are mixed, we observed that, after only a few minutes, the mixture of KAuBr_4 and α -CD leads to a pale brown co-precipitate, while all the other five mixtures remain as clear solutions in water.”

In the present manuscript, the authors are completing previous studies and demonstrate that $[\text{AuBr}_4]^-$ anion is efficiently encapsulated by $\beta\text{-CD}$ in an aqueous solutions. Moreover, brown single crystals of a 1:1 $\text{AuBr}_4 \subset \beta\text{-CD}$ complex were obtained by slowly cooling an aqueous solution and characterized using single crystal X-ray diffraction analysis. Then they revealed that upon adding a small amount of common organic solvents to an aqueous solution of the $\text{KAuBr}_4 \subset \beta\text{-CD}$ complex, brown co-precipitates form immediately. Thus, for accelerating dissolution of this kind of the host-guest complex, an antisolvent precipitation method was applied. In order to verify the generality of this antisolvent-induced co-precipitation behavior, various organic solvents were tested. Gold-recovery efficiencies ranging from 27.0 to 99.8 %, depending on the antisolvents. When a small amount of diethylene glycol dibutyl ether (DBC) was used as the antisolvent, the $[\text{AuBr}_4]^-$ anions were almost entirely precipitated from aqueous solutions with a gold-recovery

efficiency of 99.8%. The effect of the counteraction of the gold salts on gold-recovery efficiency was also investigated. Finally, the mechanism for the DBC-induced supramolecular polymerization was elucidated by X-ray crystallography and this study revealed an intriguing role of antisolvent molecules on the assembly pathway. Based on this analysis, the authors named the applied antisolvent method as an additive-induced supramolecular polymerization.

In conclusion, the reported investigations are well planned, crafted, and meticulously executed. While I do not question the validity of this work, its scientific novelty is rather limited, and I'm afraid I am not persuaded that the reported findings represent a sufficiently striking advance to justify publication in *Nat. Commun.*

Response: We appreciate the reviewer's acknowledgment of the scientific soundness of the research reported in the manuscript. We have added the suggested reference (*J. Am. Chem. Soc.* **2016**, *138*, 11643) in the revised Main Text as ref 55. We disagree, however, with the reviewer's hesitance to report our findings in *Nat. Commun.* based on a misunderstanding of our key discovery as a traditional antisolvents precipitation method. The antisolvent method is a process that involves the addition of antisolvent (a liquid that is miscible with the original solvent but shows less solubility with the desired compound) to the solution containing the desired compound. The process causes the desired compound to precipitate from the solution and form pure crystals. The principle of antisolvent relies on decreasing the solubility of the target compounds while maintaining the solubility of impurities in order to realize separation. Our discovery reported here is a very different process conceptually. We prefer to describe it as additive-induced supramolecular polymerization. Firstly, since the additives are not miscible with our solvent water, they cannot be used to adjust the solubility of compounds in water. Secondly, the additives have to participate in the self-assembly process and co-precipitate with the target compound, while antisolvents generally are not observed in the precipitates. Thirdly, antisolvent methods typically require a significant volume fraction of antisolvent to change the solubility of the desired compound. In contrast, we only need a trace amount (< 0.3% v/v) of additives to achieve a >99.8% recovery efficiency. Finally, the precipitation of the target compound is controlled by molecular encapsulation of additives rather than simply decreasing the solubility of target molecules using antisolvents. This new process is more selective and can be used to separate $[\text{AuBr}_4]^-$ from structurally similar compounds such as $[\text{PdBr}_4]^{2-}$ and $[\text{PtBr}_4]^{2-}$, which are challenging to separate by the antisolvent methods.

In order to avoid confusion and clarify the key discovery and novelty of our work compared with the previous reports, we have revised our discussion section on Page 15 of the revised Main Text as follows:

An eco-friendly and sustainable supramolecular metallurgical technology for gold recovery has been demonstrated. It is based on an additive-induced supramolecular polymerization of second-sphere coordinated adducts formed between β -cyclodextrin and tetrabromoaurate anions. The additives drive the assembly by obliging the tetrabromoaurate anions to move from the inner cavity to the primary face of two β -cyclodextrin tori, while themselves occupying the space between the secondary faces of two β -cyclodextrin tori, leading to the formation of infinite 1D supramolecular nanostructures that precipitate from the aqueous solution as cocrystals. This additive-induced supramolecular polymerization, not only provides a feasible method for the rapid crystallization of the supramolecular polymers, but it also opens the door for regulating the assembly behavior of second-sphere coordinated adducts.

In contrast to the traditional antisolvent precipitation method, the additive-induced polymerization reported in this research provides the following advantages—(i) It only requires a small volume fraction (<0.3%) of additives to precipitate effectively the target compound with a high recovery efficiency (>99.5%). (ii) The additives do not have to be mixable with the solution. (iii) The molecular recognition driven supramolecular polymerization is highly selective for the precipitation of target compounds in the presence of other structurally similar substrates.

From a practical viewpoint, this research describes a highly efficient and sustainable gold-recovery protocol. Compared with our previously reported protocol using α -cyclodextrin, the additive-induced polymerization with β -cyclodextrin comes with the following attributes—(i) The gold-recovery can be performed at a low concentration (9.3 ppm) with a much better recovery efficiency (>94%). (ii) No additional potassium ions are needed. (iii) Co-precipitation can be performed directly in acidic leaching solutions without the need of neutralization. (iv) The cost of β -cyclodextrin is lower than that of α -cyclodextrin. In summary, our establishment of additive-induced polymerization constitutes an attractive strategy for the practical recovery of gold and leads to significantly reduced energy consumption, cost inputs, and environmental pollution. We are currently optimizing the strategy to recover gold from lower-concentration gold-bearing e-waste and exploring the generality of this strategy to separate other target metal ions.

Comment 1: The authors refer in the introduction part to a few review articles on self-assemblies based on host–guest molecular recognition motifs, however they surprisingly omit those relating to CDs-based systems, e.g., Chem. Rev. 2015, 115, 15, 7240; <https://doi.org/10.1021/cr5005315>, and Chem. Rev. 2017, 117, 22, 13461; <https://doi.org/10.1021/acs.chemrev.7b00231>

Response: We thank the reviewer for bringing this literature to our attention. Accordingly, the suggested references (refs. 31 and 36) have been cited in the revised Main Text.

Reviewer comments, second round review –

Reviewer #1 (Remarks to the Author):

The authors considered and discussed all reviewers' comments. The article is now suitable for publication in its current form on Nature Communications.

Reviewer #3 (Remarks to the Author):

The authors have carefully addressed all the queries of the reviewers and performed additional and informative manipulations. I recommend the publication of this revised version

Response Letter to Reviewers' Comments

Reply to Reviewer #1

Comment: The authors considered and discussed all reviewers' comments. The article is now suitable for publication in its current form on Nature Communications.

Reply: We appreciate the referee for affirming our revision. The related comments really improve the quality of our manuscript.

Reply to Reviewer #3

Comment: The authors have carefully addressed all the queries of the reviewers and performed additional and informative manipulations. I recommend the publication of this revised version

Reply: We really thank the reviewer's approbation, which is a great encouragement for us.